**Subject Category:**
Biology (whole organism)

biotechnology/developmental biology/health and disease and epidemiology

capsaicin, osseointegration, sensory denervation, binding force

**Author for correspondence:**
Ping Gong
e-mail: dentistgong@hotmail.com

# Effects of capsaicin-induced sensory denervation on early implant osseointegration in adult rats

Bo Huang[1,2], Jun Ye[3,4], Xiaohua Zeng[5] and Ping Gong[2]

[1]State Key Laboratory of Oral Diseases, and [2]Department of Oral Implantology, West China Hospital of Stomatology, Sichuan University, Chengdu 610041, People's Republic of China
[3]Department of Prosthodontics, School and Hospital of Stomatology, Tongji University, Shanghai, People's Republic of China
[4]Engineering Research Center of Tooth Restoration and Regeneration, Shanghai, People's Republic of China
[5]Stomatology Department, The First Affiliated Hospital of Xiamen University, Xiamen, People's Republic of China

PG, 0000-0002-4782-1741

The presence of nerve endings around implants is well-known, but the interaction between the peripheral nervous system and the osseointegration of implants has not been thoroughly elucidated to date. The purpose of this study was to test the effects of selective sensory denervation on early implant osseointegration. Forty male Sprague-Dawley rats were divided randomly into two groups, group A and group B, and they were treated with capsaicin and normal saline, respectively. One week later, titanium implants were placed in the bilateral femurs of the rats. Three and six weeks after implantation, histological examination, microcomputed tomography and biomechanical testing were performed to observe the effect of sensory denervation on implant osseointegration. At three weeks and six weeks, bone area, trabecular bone volume/total bone volume and bone density were significantly lower in group A than in group B. Similarly, the bone–implant contact rate, trabecular number and trabecular thickness were clearly lower in group A than in group B at three weeks. However, the trabecular separation spacing in group A was greater than that in group B at both time points. Biomechanical testing revealed that the implant-bone binding ability of group A was significantly lower than that in group B. The research demonstrated that sensory innervation played an important role in the formation of osseointegration. Selective-sensory denervation could reduce osseointegration and lower the binding force of the bone and the implant.

# 1. Introduction

Sympathetic and sensory nerves play an important role in bone metabolism. Recently, *in vivo* and *in vitro* studies have demonstrated that the sympathetic nervous system is involved in increasing bone resorption and decreasing bone formation [1–5]. In sensory nervous regulation, a physiological interaction between sympathetic and sensory nerves occurs in osteoclastogenesis, which is based on the effects of calcitonin gene-related peptide (CGRP), a sensory neuropeptide [4,5]. An *in vitro* study showed that CGRP could induce osteoclast differentiation by isoprenaline in a mouse bone marrow cell culture system [6]. Moreover, the physiological role of the sensory nervous system in bone metabolism was demonstrated *in vivo* [5]. A study showed that high-dose (150 mg kg$^{-1}$) capsaicin treatment reduced trabecular bone volume (BV/TV) due to increased trabecular separation (Tb.Sp) in the proximal tibia and the modification of mechanical properties, such as strength, ductility and toughness, towards increasing bone fragility by promoting the differentiation and maturation of osteoclasts [7]. Studies have indicated that nerve tissue in bone is involved in bone remodelling, primarily by affecting bone cell metabolism and regulating blood flow changes in the bone [8,9]. Some studies also showed that there was an association between the distribution of nerve endings in bone tissue and the activity of bone reconstruction because there were more nerve fibres in areas where bone remodelling was active [10,11].

Calcitonin gene-related peptide (CGRP), a neuropeptide synthesized in sensory neurons in the spinal root ganglion that primarily exists in the C-type and A-type sensory nerve endings, is transferred to sensory nerve endings through the axon reversely and can promote bone formation and inhibit bone resorption [12]. CGRP was also considered to be a potent microvasodilator and a regulator of some actions of the sympathetic nervous system [13,14]. Similar to CGRP, substance P (SP) is also a nociceptive neuropeptide associated with unmyelinated C-type fibres. Although SP often coexists with CGRP, the effect of SP was weaker than that of CGRP in promoting bone formation and inhibiting bone resorption [15].

Capsaicin, a neurotoxic agent, can activate transient receptor potential vanilloid 1 (TRPV1), which is expressed by most unmyelinated sensory neurons and some small-diameter myelinated sensory neurons [16,17]. The activation of TRPV1 could lead to neurotoxicity by gathering calcium and sodium cations [18]. With low doses of capsaicin, TRPV1 activation plays an important role in the sensation of pain. However, at high doses, capsaicin acts as an excitatory neurotoxin, leading to the overactivation of TRPV1 and inducing cell death by massive ion influx [18,19]. In the neurons of the dorsal root ganglion [19], high doses of capsaicin could lead to a loss of the immunoreactivities of CGRP and SP [20,21]. Systemic capsaicin treatment exerted a long-term blocking action against the sensory receptor, which could be used in the investigation of the function of afferent neurons [7].

It has been reported that there are neurofilament protein (NFP)-positive nerve endings in the bone within 200 μm extending around the implant surface [22]. Nevertheless, the function of these nerve fibres is still unclear [22]. Whether these nerve fibres serve as peripheral receptors or play a regulatory role in bone metabolism, especially in the formation and maintenance of implant osseointegration, has drawn considerable attention from researchers.

According to previous studies and the mechanism of osseointegration, we hypothesize that the nerve distribution around the implant is associated with the formation and maintenance of osseointegration and plays an essential role in the metabolic activity of peri-implant bone tissue [22]. Therefore, in this study, we aimed to investigate the effects of capsaicin-induced sensory denervation on implant osseointegration.

# 2. Material and methods

## 2.1. Animals

The animal experiment protocol was approved by the Animal Research Ethics Committee of West China Hospital of Stomatology, Sichuan University (no. WCCSIRB-D-2015026). A total of 40 adult male Sprague-Dawley rats, weighing 300–350 kg, were divided into two groups of 20. The rats were kept in the State National Key Laboratory of Biotherapy.

## 2.2. Capsaicin treatment

Capsaicin (Sigma, St Louis, MO, USA) and vehicle (10 : 10 : 80 v/v of Tween-80, ethanol, saline) were freshly prepared before treatment each day. The capsaicin was sonicated in vehicle until

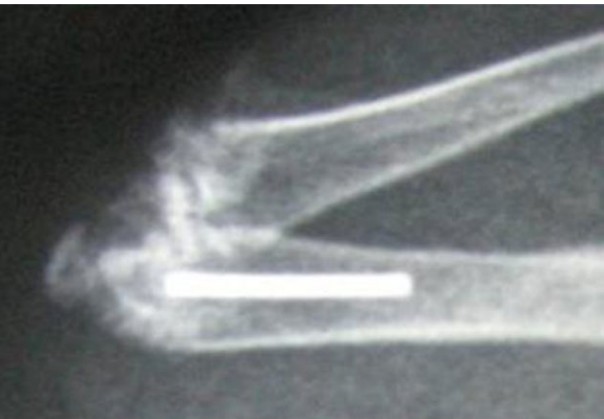

**Figure 1.** Radiograph of femur with implant.

homogeneously suspended. Before the injection, the rats were anaesthetized with 10% chloral hydrate. Then, the experimental group was injected subcutaneously above the femur with capsaicin, and the sensory-intact group was injected with vehicle. A 30-gauge needle was inserted retrograde into the anterior aspect of the thigh parallel to the femur and slid along its anterior cortex. As the needle was withdrawn, 125 µl of 1% capsaicin or vehicle was injected along the anterior cortex. This procedure was repeated for the posterior cortex. All treated rats were subjected to implant surgery one week later.

## 2.3. Implant preparation

The rod-shaped implants used in this study were made of commercially pure titanium and supplied by the National Engineering Research Center of Biomaterials (Sichuan University, China). The implants measured 10 mm in length and 1 mm in diameter. All implants were machined and grit-blasted with 25 µm aluminium oxide ($Al_2O_3$) particles and then sequentially washed with 40% (v/v) NaOH and deionized water in an ultrasonic bath. Before the implant surgery, all implants were sterilized in an autoclave.

## 2.4. Implant surgery

All rats were anaesthetized with intraperitoneal injections of 3% pentobarbital sodium (4 mg/100 g) and then subjected to implant placement. Eighty implants were used, and one implant was used in the left leg and one in the right leg for each rat. Briefly, the implant sites were prepared on both sides of the medial side of the knee joint surfaces of the femurs by sequential drilling under cooled sterile saline irrigation with 0.7 and 1.0 mm surgical stainless-steel twist drills. Then, the implants were press-fitted into the prepared holes. After implant insertion, the muscles were carefully sutured, which protected the implant from the surrounding environment and further guaranteed the stability of the implant. All rats received an intramuscular antibiotic injection for 3 days after the operation and were allowed free movement without any restriction. A representative radiograph of the femur with the implant is shown in figure 1. Ten animals were killed at each time point in each group.

## 2.5. Histomorphometric measurement

Animals were sacrificed three and six weeks after implantation, and then the femur with implants was isolated immediately with the soft tissue removed. The femurs were fixed in 4% paraformaldehyde for 48 h and decalcified in 20% EDTA (pH 7.4) for four weeks; then, the specimens were embedded in paraffin, sectioned longitudinally at 5 µm thickness with a Leica RM 2255 microtome (Leica, Germany) and stained with HE. The other isolated femurs were maintained in a 4% neutral formalin buffered solution for 2 days and were washed and dehydrated through a graded series (60%–80%–90%–100%) of ethanol solutions. These specimens were embedded in methylmethacrylate without decalcification [23]. Cross-sections through the horizontal axis of implants were prepared using a rotary diamond saw (SP1600/2600, Leica) and stained with 1% toluidine blue. Histomorphometric analysis was performed on sections with a Leica DMI 6000 B microsystem (Leica). Bone-to-implant

contact (BIC) was measured approximately 2 mm above the epiphyseal plate, and bone area (BA) was measured around the area extending 200 μm from the implant surface [24].

## 2.6. Microscopic computerized tomography evaluation

The femurs with implants ($n = 10$ specimens per group) were scanned on a microscopic computerized tomography (micro-CT) system (micro-CT 80 scanner Scanco Medical, Bassersdorf, Switzerland), which was set to 90 kVp, 200 μA, 18 W and 500 ms integration time. A multi-level threshold procedure was applied to discriminate bone from other tissues. The three-dimensional images reconstructed from micro-tomographic slices were used for qualitative and quantitative evaluation with the constrained three-dimensional Gaussian filter ($\sigma = 1.2$, support = 1) for partial suppression of the noise in the volumes [25]. The volume of interest (VOI) was defined as bone tissues from 2.0 mm above the growth plate to proximal 50 slices extending with a radius of 200 μm from the implant surface. After segmentation, the bone volume per total volume (BV/TV), the mean trabecular thickness (Tb.Th), the mean trabecular number (Tb.N), the mean trabecular separation (Tb.Sp), and the mean connective density (Conn.D) were assessed within the VOI zone. The osseointegration ratio (OI%) was calculated as the ratio between bone and total voxels in direct contact with the implant.

## 2.7. Biomechanical test

Immediately after micro-CT evaluation, a push-out test was performed on four specimens per group using a universal material testing system (Instron 5566; Instron, Norwood, MA, USA). A custom-designed holder was used to maintain the downward compression to centre the implant and align it vertically. Approximately 2 mm of the implant end in the femur metaphysis was exposed by epiphyseal separation. The compression speed was 1 mm min$^{-1}$. The maximal force and interfacial shear strength were calculated by the displacement versus force.

## 2.8. Statistical analysis

All experiments were performed in quadruplicate and reproduced at least four separate times. Statistical analysis of the data was performed with SPSS 17.0 (Chicago, IL) using one-way ANOVA to compare the means of all groups followed by Newman–Keuls *post hoc* test. The results were considered to be significant if the two-tailed $p$ value was less than 0.05.

# 3. Results

## 3.1. Histomorphometric analysis

The decalcified sections showed that the trabecular number in the cortical bone of the distal femoral metaphysis was lower in the sensory-denervated group than in the sensory-intact group three weeks after implantation (figure 2). Six weeks after implantation, the trabecular number in the two groups increased. However, the number in the sensory-denervated group was still less than that in the sensory-intact group.

## 3.2. BA and BIC analysis

Implant osseointegration and peri-implant bone mass are shown in figure 3*a*. The results from the histomorphometric analysis are presented as BIC and BA (figure 3*b,c*). Three weeks after implantation, the BA and BIC in the sensory-denervated group were 15.76% ± 1.78% and 45.10% ± 3.62%, respectively. In the sensory-intact group, the BA and BIC increased approximately 32% and 14%, respectively, compared to those values in the sensory-denervated group ($p < 0.05$). Six weeks after implantation, the BA and BIC in the sensory-denervated group were 28.39% ± 3.07% and 66.30% ± 5.89%, respectively. The BA in the sensory-intact group increased to more than 20% higher than the BA in the sensory-denervated group ($p < 0.05$). However, there was no significant difference in the BIC between the sensory-denervated group and the sensory-intact group at six weeks after implantation ($p > 0.05$).

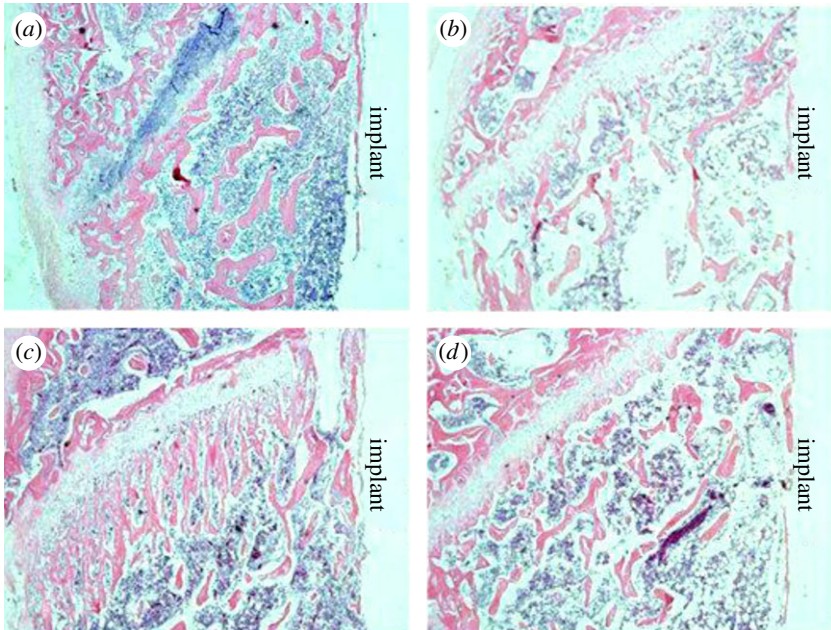

**Figure 2.** Histological images of the distal femur with implants three and six weeks after implantation. (*a*) sensory-intact group at three weeks after implantation; (*b*) sensory-denervated group at three weeks after implantation; (*c*) sensory-intact group at six weeks after implantation; (*d*) sensory-denervated group at six weeks after implantation; HE staining, original magnification ×40. $N = 4$.

## 3.3. Micro-CT evaluation

Three-dimensional micro-CT images depicted the bone–implant interface and trabecular topography among rats subjected to sensory denervation or not in figure 4*a*. The quantitative evaluation provided more detailed information on the OI% and trabecular parameters around the implants, as shown in figure 4*b*. Three weeks after implantation, the BV/TV, Conn.D, Tb.N, Tb.Th and OI% in the sensory-intact group increased to approximately 30%, 40%, 20%, 25% and 15% higher than those values in the sensory-denervated group, respectively ($p < 0.05$). However, TB.Sp decreased to approximately 15% lower in the sensory-intact group than in the sensory-denervated group. At six weeks, the BV/TV and Conn.D in the sensory-intact group increased to approximately 15% and 11% higher than those values in the sensory-denervated group, respectively ($p < 0.05$). There were no significant differences in the Tb.N, Tb.Th, Tb.Sp and OI% values between the sensory-denervated group and the sensory-intact group at six weeks after implantation ($p > 0.05$).

## 3.4. Biomechanical test

The results of the push-out test were used to examine the maximal push-out force and ultimate shear strength in figure 5. Three and six weeks after implantation, the values in the two biomechanical indices were lower in the sensory-denervation group than in the sensory-intact group. At three weeks, the maximal push-out force and the ultimate shear strength in the sensory-denervated group were approximately 20 N and 0.75 N mm$^{-2}$, respectively. These values increased to 38 N and 1.51 N mm$^{-2}$ in the sensory-intact group. At six weeks, the maximal push-out force in the sensory-denervated group and the sensory-intact group increased to 72 N and 94 N, respectively, and the difference was statistically significant ($p < 0.05$). The results for the ultimate shear strength were similar to those for the maximal push-out force.

## 4. Discussion

Many clinical studies and fundamental studies have suggested that the peripheral nervous system could regulate bone metabolism to a certain extent [26–30]. Most previous studies used the method of nerve trunk cutting to study the effect of denervation on bone metabolism [31,32]. However, the nerve trunk is commonly composed of various nerves, including the motor, sensory and/or autonomic nerves,

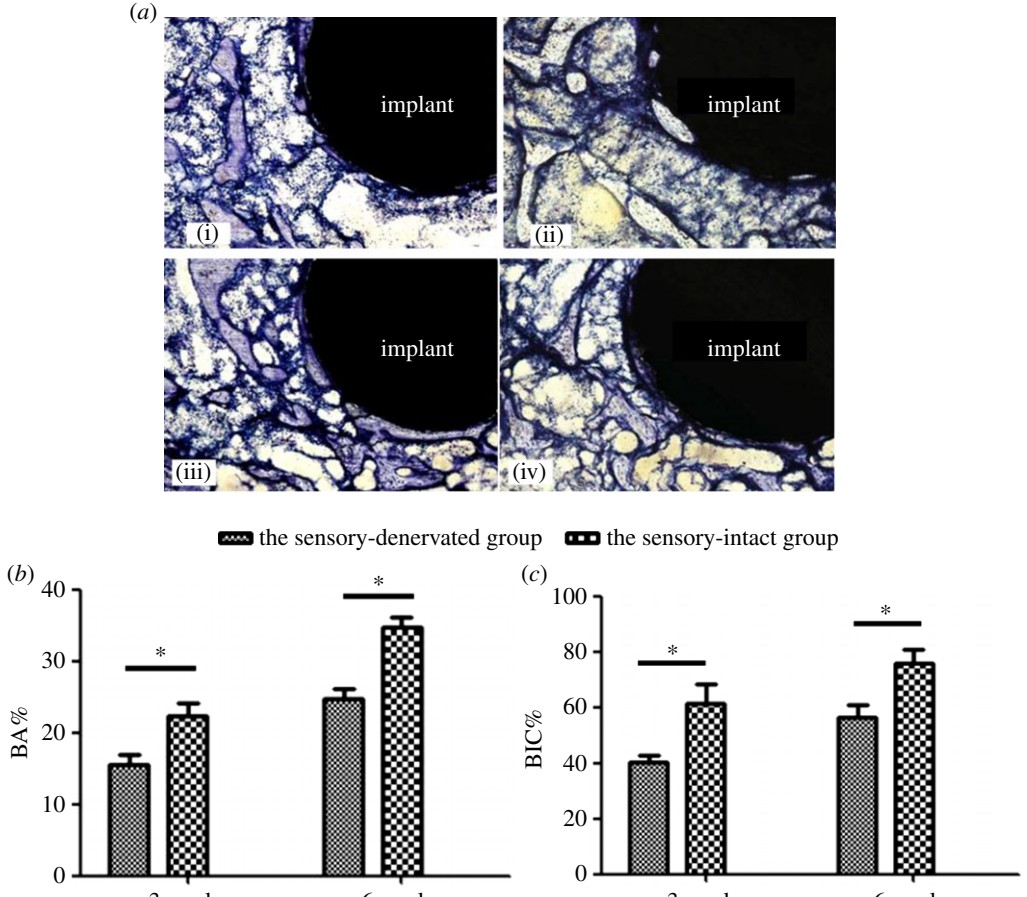

**Figure 3.** (a) Toluidine blue staining, BA and BIC analysis. (i) Sensory-intact group three weeks after implantation; (ii) sensory-denervated group three weeks after implantation; (iii) sensory-intact group six weeks after implantation; (iv) sensory-denervated group six weeks after implantation; toluidine blue staining, original magnification ×40. N = 4. (b) Histograms of the BA in the histomorphometric analysis (data are expressed as the mean ± s.d.; N = 4 specimens/group. *p < 0.05). (c) Histograms of the BIC in the histomorphometric analysis (data are expressed as the mean ± s.d.; N = 4 specimens/group. *p < 0.05).

and this method cannot distinguish the effects of different nerve fibres that may lead to changes in bone tissue [31,32].

Capsaicin, a neurotoxic agent, could specifically denervate the sensory nerves of CGRP-positive and SP-positive patients [29]. The model of sensory denervation could be built by submucosal injection of capsaicin at the spinal cord or local injection of capsaicin [28]. By submucosal injection at the spinal cord, capsaicin could be widely distributed in neurons expressing TRPV1 in animals' airways, which can lead to neurogenic lung inflammation [20,21,33,34]. The mortality of acute pulmonary oedema caused by lung neurogenic inflammation is very high; therefore, local injection of capsaicin at the proposed denervated tissues has been recommended in many studies [20,21,33,34].

In this study, we denervated the sensory nerve by subcutaneously injecting capsaicin above the femur. One week later, the capsaicin had taken effect, and selective lesions in the sensory fibres occurred. Then, the implantation was performed. The results of this study showed that the trabecular number and CGRP-IR-positive fibres in the distal femoral metaphysis in the sensory-denervated group were lower than those in the sensory-intact group three weeks after implantation. Neurotrophins, such as nerve growth factor (NGF), secreted by the local bone could induce the reinnervation of other lower extremity sensory neurons; thus, the trabecular number and CGRP-IR-positive fibres in the sensory-denervated group were increased at six weeks compared to three weeks after implantation. Although neurotrophin increased at six weeks, the trabecular number and CGRP-IR-positive fibres in the sensory-denervated group were still lower than those in the sensory-intact group. The results described above demonstrated that this approach was an effective method of sensory denervation by subcutaneous injection of capsaicin.

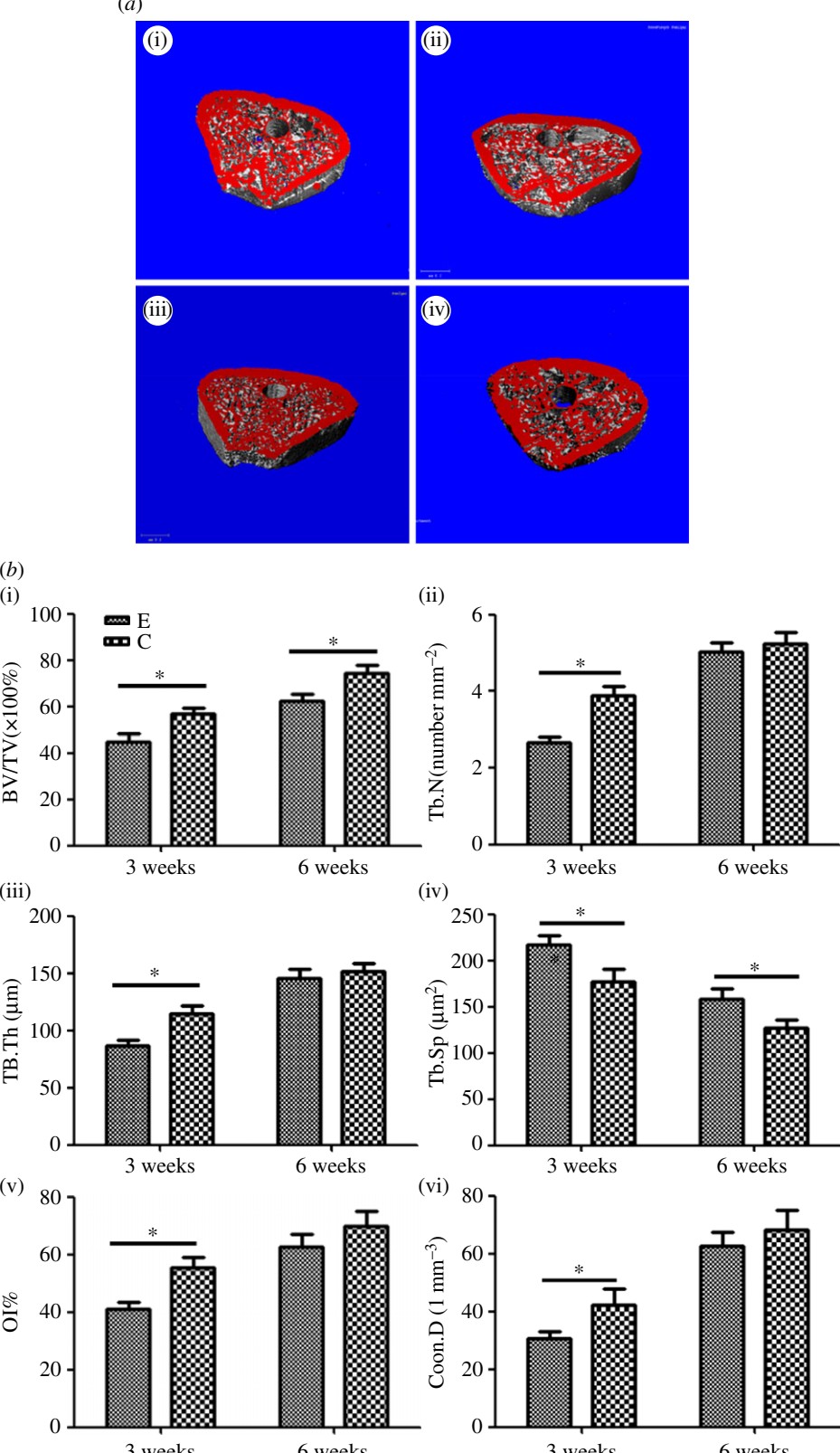

**Figure 4.** (*a*) Micro-CT images (transverse 3-D images through a cross-sectional plane of the implants) of distal femurs with implants three and six weeks after implantation. (i) sensory-intact group at three weeks after implantation; (ii) sensory-denervated group at three weeks after implantation; (iii) sensory-intact group at six weeks after implantation; (iv) sensory-denervated group at six weeks after implantation. (*b*) Quantitative results of the micro-CT evaluation three and six weeks after implantation. 3C: sensory-intact group at three weeks after implantation; 3E: sensory-denervated group at three weeks after implantation; 6C: sensory-intact group at six weeks after implantation; 6E: sensory-denervated group at six weeks after implantation (data are expressed as the mean $\pm$ s.d.; $N = 4$ specimens/group. *$p < 0.05$).

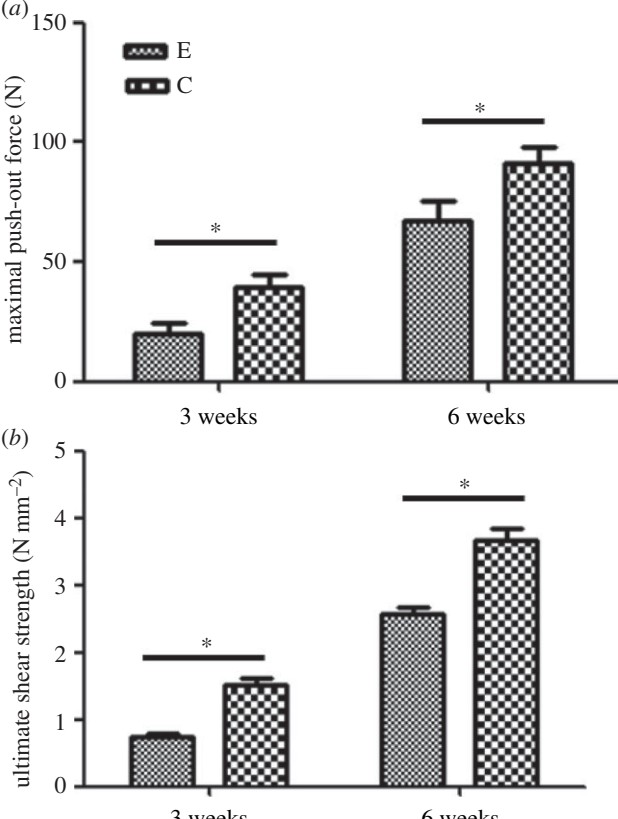

**Figure 5.** Histograms of the mechanical push-out test parameters three and six weeks after implantation. (*a*) maximal push-out force, (*b*) ultimate shear strength (data are expressed as the mean $\pm$ s.d.; $N = 4$ specimens/group. *$p < 0.05$).

Offley *et al.* reported that sensory neurons have efferent functions in remodelling bone, and this process was mediated by transmitters that were released from the peripheral nerve terminals [21]. When treated with capsaicin, the osteoclast number would increase, but the osteoblast activity, osteoblast number, bone formation, trabecular bone volume and femoral strength would decrease significantly [21]. However, another study showed that low-dose (37.5 mg kg$^{-1}$) capsaicin had no influence on bone remodelling. High-dose capsaicin (150 mg kg$^{-1}$) significantly increased the level of tartrate-resistant acid phosphatase form 5b (TRAP 5b) in the plasma but had no influence on the plasma osteocalcin concentration, suggesting that sensory nerve innervation contributed to the maintenance of trabecular bone mass and its mechanical properties by inhibiting bone resorption [7,21]. Similar to our research, these results indicated that the sensory nerve played an important role in the maintenance of trabecular bone mass.

Osseointegration refers to the placement of new bone deposits on the surface of the implant directly without fibres, forming a combination between them [35]. There is a close relationship between osseointegration and peri-implant bone metabolism. Three weeks after implantation, the BIC and BA of the sensory-denervated group were 13% and 25% less than those of the sensory-intact group, respectively. The difference in the BIC between the denervation group and the sensory-intact group was not statistically significant. It might be that the bone matrix secreted by osteoblasts gathers and adheres to the implant surface, which would be beneficial to the formation of bone on the surface of the implant in the early stage. In the repair process, the osteogenesis on the surface of the implant, also known as the BIC, was decreased significantly because sensory denervation could decrease the activity of osteoblasts and bone formation [7,21]. The results of the micro-CT evaluation also revealed the adverse effects of sensory denervation on implant osseointegration. The BV/TV, Conn.D, Tb.N, Tb.Th and OI% of the sensory-intact group were higher than those in the sensory-denervated group. These differences illustrated that sensory denervation could decrease bone formation and mineralization around the implants. The maximal push-out force of the sensory-denervated group was decreased by approximately 50% compared to the sensory-intact group, which means that sensory denervation could reduce the resulting strength of the implant.

Similar to the results of previous studies, our research showed that there was bidirectional regulation between the nerve and bone tissue [7,21,25–27,36]. During bone remodelling, neurotrophins, such as NGF and CGRP, secreted by the local bone can induce the reinnervation of other lower extremity sensory neurons, which will regulate the metabolism of bone tissue [36,37]. Conversely, neurotrophin could promote angiogenesis, which would benefit the reconstruction of bone around the implant [30,37,38].

# 5. Conclusion

Selective sensory denervation treatment impaired implant osseointegration, and this treatment decreased the quality and quantity of the bone around the implant, weakening the force of the implant, which suggested that sensory innervation plays an important role in the formation and maintenance of implant osseointegration.

Ethics. The study was reviewed and approved by the Animal Research Ethics Committee of West China Hospital of Stomatology, Sichuan University (no. WCCSIRB-D-2015026).

Data accessibility. Our data are deposited at Dryad Digital Repository: http://dx.doi.org/10.5061/dryad.k53t495 [39].

Authors' contributions. B.H. and J.Y. carried out the molecular laboratory work, participated in the data analysis, carried out sequence alignments, participated in the design of the study and drafted the manuscript; X.Z. carried out the statistical analyses; B.H. collected the field data; P.G. conceived of the study, designed the study, coordinated the study and helped draft the manuscript. All authors gave final approval for publication.

Competing interests. The authors declare no competing interests.

Funding. This work was supported by the National Natural Science Foundation of China (nos. 813009061 and 81571008).

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
