## [Reviewer comments · Royal Society Open Science]

Review History

RSOS-181082.R0 (Original submission)

Review form: Reviewer 1

Is the manuscript scientifically sound in its present form?

Yes

Are the interpretations and conclusions justified by the results?

Yes

Is the language acceptable?

Yes

Is it clear how to access all supporting data?

Yes

Do you have any ethical concerns with this paper?

No

Have you any concerns about statistical analyses in this paper?

No

Recommendation?

Major revision is needed (please make suggestions in comments)

Comments to the Author(s)

1. Glad to read this manuscript which try to induce nerve injury by using capsaicin, and then observe the effects on peri-implant osseointegration. The positive results confirmed this unique hypothesis.
2. The methods and materials need to be refined further. There were one experiment group and one control, why not add another group with capsaicin but without implant placement? Should not this group add more proof to the conclusion?
3. The microCT scanning parts is not clear. What kind of energy filter was used for scanning this high dense implant inside the bone tissue? And what is the basic imaging processing dealing with metal artefacts and beam hardening?
4. The first image in the histological graph was a bit dark compared to other ones.
5. In the discussion part, " One week later, the corresponding sensory neurons were completely lost", No reference? no info about how to examine.
6. In the discussion part, "Neurotrophin such as NGF secreted by the local bone could induce the reinnervation of other lower extremity sensory neurons, so the trabecular number and the positive fibers of CGRP-IR of the sensory-denervated group were increased at six weeks compared to three weeks after implantation. Although Neurotrophin increased at six weeks, the trabecular number and the positive fibers of CGRP-IR in the sensory-denervated group still less than that in the sensory-intact group." positive fibers of CGRP-IR was not clear in the previous results part.
7. "Offley at al reported " should be " offley et al. reported".
8. "On the other hand, the Neurotrophin could promote angiogenesis, which would benefit the reconstruction of bone around the implant [38,40]." is this a clear conclusion or just an inference ?

Review form: Reviewer 2 (Liang Zhang)

Is the manuscript scientifically sound in its present form?

Yes

Are the interpretations and conclusions justified by the results?

Yes

Is the language acceptable?

Yes

Is it clear how to access all supporting data?

Yes

Do you have any ethical concerns with this paper?

No

Have you any concerns about statistical analyses in this paper?

Yes

Recommendation?

Accept with minor revision (please list in comments)

Comments to the Author(s)

Please clarify the following questions:

1. There are flaws in language which needs further proofreading and correction.
2. The authors claim that one week after the injection of capsaicin, the sensory neurons were completely destroyed, for which no tests were carried however. This is theoretically true only when the procedure is properly done and thus a test of sensory function before the placement of implants would be helpful to confirm the sensory denervation.
3. The grouping and time points of examinations were not clearly described. It was only stated in line 6 of page 3 that there are 40 rats which were divided into two groups. The authors however did several tests with the samples drawn from both femurs of all animals at two time points after the surgery. It was not clarified how many samples from either group underwent each of the examinations, and it is thus not possible to appreciate the correctness of statistics.

Liang Zhang, DDS, PhD

Department of Implant Dentistry

West China School and Hospital of Stomatology, Sichuan University

Decision letter (RSOS-181082.R0)

24-Sep-2018

Dear Professor Gong,

The editors assigned to your paper ("Effects of capsaicin-induced sensory denervation on early implant osseointegration in adult rats") have now received comments from reviewers. We would like you to revise your paper in accordance with the referee and Associate Editor suggestions which can be found below (not including confidential reports to the Editor). Please note this decision does not guarantee eventual acceptance.

Please submit a copy of your revised paper before 17-Oct-2018. Please note that the revision deadline will expire at 00.00am on this date. If we do not hear from you within this time then it will be assumed that the paper has been withdrawn. In exceptional circumstances, extensions may be possible if agreed with the Editorial Office in advance. We do not allow multiple rounds of revision so we urge you to make every effort to fully address all of the comments at this stage. If deemed necessary by the Editors, your manuscript will be sent back to one or more of the original reviewers for assessment. If the original reviewers are not available, we may invite new reviewers.

When submitting your revised manuscript, you must respond to the comments made by the referees and upload a file "Response to Referees" in "Section 6 - File Upload". Please use this to document how you have responded to the comments, and the adjustments you have made. In

order to expedite the processing of the revised manuscript, please be as specific as possible in your response.

- Data accessibility

If you wish to submit your supporting data or code to Dryad (<http://datadryad.org/>), or modify your current submission to dryad, please use the following link:
<http://datadryad.org/submit?journalID=RSOS&manu=RSOS-181082>

- Competing interests

- Authors' contributions

- Acknowledgements

- Funding statement

Please note that Royal Society Open Science charge article processing charges for all new submissions that are accepted for publication. Charges will also apply to papers transferred to Royal Society Open Science from other Royal Society Publishing journals, as well as papers submitted as part of our collaboration with the Royal Society of Chemistry (<http://rsos.royalsocietypublishing.org/chemistry>). If your manuscript is newly submitted and subsequently accepted for publication, you will be asked to pay the article processing charge, unless you request a waiver and this is approved by Royal Society Publishing. You can find out more about the charges at <http://rsos.royalsocietypublishing.org/page/charges>. Should you have any queries, please contact openscience@royalsociety.org.

on behalf of Prof. Kevin Padian (Subject Editor)
openscience@royalsociety.org

Subject Editor's comments:

I believe that this paper is more relevant to clinical studies than to basic biological research; therefore I would like the authors to address more, in their conclusions, what relevance their work has for basic biological research.

In addition, unfortunately the English (although very good) is not up to publishable standards. Please have a native speaker of English edit your next version, otherwise we will be unable to consider it further. Thanks for your attention to this.

Associate Editor's comments:

Please ensure that you not only address the scientific concerns of the referees, but seek advice on polishing the English language, as recommended by the referees. You can find a range of services to assist with this at <https://royalsociety.org/journals/authors/language-polishing/>.

Comments to Author:

Reviewers' Comments to Author:

Reviewer: 1

Comments to the Author(s)

1. Glad to read this manuscript which try to induce nerve injury by using capsaicin, and then observe the effects on peri-implant osseointegration. The positive results confirmed this unique hypothesis.
2. The methods and materials need to be refined further. There were one experiment group and one control, why not add another group with capsaicin but without implant placement? Should not this group add more proof to the conclusion?

3. The microCT scanning parts is not clear. What kind of energy filter was used for scanning this high dense implant inside the bone tissue? And what is the basic imaging processing dealing with metal artefacts and beam hardening?
4. The first image in the histological graph was a bit dark compared to other ones.
5. In the discussion part, " One week later, the corresponding sensory neurons were completely lost", No reference? no info about how to examine.
6. In the discussion part, "Neurotrophin such as NGF secreted by the local bone could induce the reinnervation of other lower extremity sensory neurons, so the trabecular number and the positive fibers of CGRP-IR of the sensory-denervated group were increased at six weeks compared to three weeks after implantation. Although Neurotrophin increased at six weeks, the trabecular number and the positive fibers of CGRP-IR in the sensory-denervated group still less than that in the sensory-intact group." positive fibers of CGRP-IR was not clear in the previous results part.
7. "Offley at al reported " should be " offley et al. reported".
8. "On the other hand, the Neurotrophin could promote angiogenesis, which would benefit the reconstruction of bone around the implant [38,40]." is this a clear conclusion or just an inference ?

Reviewer: 2

Comments to the Author(s)

Please clarify the following questions:

1. There are flaws in language which needs further proofreading and correction.
2. The authors claim that one week after the injection of capsaicin, the sensory neurons were completely destroyed, for which no tests were carried however. This is theoretically true only when the procedure is properly done and thus a test of sensory function before the placement of implants would be helpful to confirm the sensory denervation.
3. The grouping and time points of examinations were not clearly described. It was only stated in line 6 of page 3 that there are 40 rats which were divided into two groups. The authors however did several tests with the samples drawn from both femurs of all animals at two time points after the surgery. It was not clarified how many samples from either group underwent each of the examinations, and it is thus not possible to appreciate the correctness of statistics.

Liang Zhang, DDS, PhD

Department of Implant Dentistry

West China School and Hospital of Stomatology, Sichuan University

Author's Response to Decision Letter for (RSOS-181082.R0)

See Appendix A.

RSOS-181082.R1 (Revision)

Review form: Reviewer 1

Is the manuscript scientifically sound in its present form?

Yes

Are the interpretations and conclusions justified by the results?

Yes

Is the language acceptable?

Yes

Is it clear how to access all supporting data?

Yes

Do you have any ethical concerns with this paper?

No

Have you any concerns about statistical analyses in this paper?

No

Recommendation?

Accept with minor revision (please list in comments)

Comments to the Author(s)

Thank you for your further work, which makes sense and contributes to the final paper! Please note that it is better to use same term in one paper in stead of using many different ones, eg., micro-CT, μ CT, μ -CT, micro computed tomography.

Review form: Reviewer 2 (Liang Zhang)

Is the manuscript scientifically sound in its present form?

Yes

Are the interpretations and conclusions justified by the results?

Yes

Is the language acceptable?

Yes

Is it clear how to access all supporting data?

Yes

Do you have any ethical concerns with this paper?

No

Have you any concerns about statistical analyses in this paper?

No

Recommendation?

Accept as is

Comments to the Author(s)

Thanks for the answer to the questions I've raised previously.

Decision letter (RSOS-181082.R1)

13-Nov-2018

Dear Professor Gong:

On behalf of the Editors, I am pleased to inform you that your Manuscript RSOS-181082.R1 entitled "Effects of capsaicin-induced sensory denervation on early implant osseointegration in adult rats" has been accepted for publication in Royal Society Open Science subject to minor revision in accordance with the referee suggestions. Please find the referees' comments at the end of this email.

The reviewers and Subject Editor have recommended publication, but also suggest some minor revisions to your manuscript. Therefore, I invite you to respond to the comments and revise your manuscript.

- Ethics statement

- Data accessibility

<http://datadryad.org/submit?journalID=RSOS&manu=RSOS-181082.R1>

- Competing interests

- Authors' contributions

- Acknowledgements

- Funding statement

Because the schedule for publication is very tight, it is a condition of publication that you submit the revised version of your manuscript before 22-Nov-2018. Please note that the revision deadline will expire at 00.00am on this date. If you do not think you will be able to meet this date please let me know immediately.

- 1) A text file of the manuscript (tex, txt, rtf, docx or doc), references, tables (including captions) and figure captions. Do not upload a PDF as your "Main Document".
- 2) A separate electronic file of each figure (EPS or print-quality PDF preferred (either format should be produced directly from original creation package), or original software format)

- 3) Included a 100 word media summary of your paper when requested at submission. Please ensure you have entered correct contact details (email, institution and telephone) in your user account
- 4) Included the raw data to support the claims made in your paper. You can either include your data as electronic supplementary material or upload to a repository and include the relevant doi within your manuscript
- 5) All supplementary materials accompanying an accepted article will be treated as in their final form. Note that the Royal Society will neither edit nor typeset supplementary material and it will be hosted as provided. Please ensure that the supplementary material includes the paper details where possible (authors, article title, journal name).

Please note that Royal Society Open Science charge article processing charges for all new submissions that are accepted for publication. Charges will also apply to papers transferred to Royal Society Open Science from other Royal Society Publishing journals, as well as papers submitted as part of our collaboration with the Royal Society of Chemistry (<http://rsos.royalsocietypublishing.org/chemistry>). If your manuscript is newly submitted and subsequently accepted for publication, you will be asked to pay the article processing charge, unless you request a waiver and this is approved by Royal Society Publishing. You can find out more about the charges at <http://rsos.royalsocietypublishing.org/page/charges>. Should you have any queries, please contact openscience@royalsociety.org.

on behalf of Professor Kevin Padian (Subject Editor)
openscience@royalsociety.org

Associate Editor Comments to Author:

Thank you for submitting your revision. Overall, the reviewers are pleased with the efforts you have made; however, there is a minor outstanding suggestion to aid clarity: per the suggestion of one of the referees, please ensure you are consistent in your use of terminology. We look forward to receiving the revision.

Reviewer comments to Author:

Reviewer: 2

Comments to the Author(s)

Thanks for the answer to the questions I've raised previously.

Reviewer: 1

Comments to the Author(s)

Thank you for your further work, which makes sense and contributes to the final paper! Please note that it is better to use same term in one paper in stead of using many different ones, eg., micro-CT, μ CT, μ -CT, micro computed tomography.

Author's Response to Decision Letter for (RSOS-181082.R1)

See Appendix B.

Decision letter (RSOS-181082.R2)

20-Nov-2018

Dear Professor Gong,

I am pleased to inform you that your manuscript entitled "Effects of capsaicin-induced sensory denervation on early implant osseointegration in adult rats" is now accepted for publication in Royal Society Open Science.

on behalf of Prof Kevin Padian (Subject Editor)
openscience@royalsociety.org

Appendix A

Answer: The animal ethics committee (Research Ethics Committee of West China Hospital of Stomatology) approved our experiment. The number is WCCSIRB-D-2014-65. And we have attached the letter.

- Data accessibility

Answer: We have uploaded the data to the website. And we have added it at the "Data accessibility" in the manuscript

<https://datadryad.org/review?doi=doi:10.5061/dryad.k53t495>

DOI: doi:10.5061/dryad.k53t495

- Competing interests

Answer: We promise that we declare we have no competing interests. We have added it in the "Competing interests" in the manuscript.

- Authors' contributions

Answer: Bo Huang and Jun Ye carried out the molecular lab work, participated in data analysis, carried out sequence alignments, participated in the design of the study and drafted the manuscript; Xiaohua Zeng carried out the statistical analyses; Bo Huang collected field data; Ping Gong conceived of the study, designed the study, coordinated the study and helped draft the manuscript. All authors gave final approval for publication.

And we have added it in the "Authors' contributions" in the manuscript.

- Acknowledgements

Answer: All the people who contributed to the study have meet the authorship criteria, and they are all listed in the article as the authorship.

And we have added it in the manuscript.

- Funding statement

Answer: This work was supported by the National Natural Science Foundation of China (NO.813009061 and NO.81571008).

And we have added it in the manuscript.

Please note that Royal Society Open Science charge article processing charges for all new submissions that are accepted for publication. Charges will also apply to papers transferred to Royal Society Open Science from other Royal Society Publishing journals, as well as papers submitted as part of our collaboration with the Royal Society of Chemistry

(<http://rsos.royalsocietypublishing.org/chemistry>). If your manuscript is newly submitted and subsequently accepted for publication, you will be asked to pay the article processing charge, unless you request a waiver and this is approved by Royal Society Publishing. You can find out more about the charges at <http://rsos.royalsocietypublishing.org/page/charges>. Should you have any queries, please contact openscience@royalsociety.org.

on behalf of Prof. Kevin Padian (Subject Editor)
openscience@royalsociety.org

Subject Editor's comments:

I believe that this paper is more relevant to clinical studies than to basic biological research; therefore I would like the authors to address more, in their conclusions, what relevance their work has for basic biological research.

Answer: Thank you for your suggestions. We have addressed the relationship between our work and basic biological research in the conclusion.

In addition, unfortunately the English (although very good) is not up to publishable standards. Please have a native speaker of English edit your next version, otherwise we will be unable to consider it further. Thanks for your attention to this.

Answer: We are not native speakers of English, so it is a little hard for us to have a well written in the first time. But we have re-edited the language one sentence by one sentence. After that, we have invited the professionals (Recommended by Associate Editor) to modify the language of the manuscript. Thank you for your suggestions.

Associate Editor's comments:

Please ensure that you not only address the scientific concerns of the referees, but seek advice on polishing the English language, as recommended by the referees. You can find a range of services to assist with this at <https://royalsociety.org/journals/authors/language-polishing/>.

Answer:

Thank you for your suggestions, we find a services (AJE, American Journal Experts) to help me modify the language in the <https://royalsociety.org/journals/authors/language-polishing/>.

Comments to Author:

Reviewers' Comments to Author:

Reviewer: 1

Comments to the Author(s)

1. Glad to read this manuscript which try to induce nerve injury by using capsaicin, and then observe the effects on peri-implant osseointegration. The positive results confirmed this unique hypothesis.

Answer: Yes, in this manuscript, we found capsaicin could injure the nerve, and then affect the peri-implant osseointegration.

2. The methods and materials need to be refined further. There were one experiment group and one control, why not add another group with capsaicin but without implant placement? Should not this group add more proof to the conclusion?

Answer: Previous studies have showed that capsaicin-sensitive sensory neurons play an important role in bone modeling and it could result in the reduction of bone density and mechanical properties. The trabecular bone volume was reduced by 40% and the ultimate strength in decreased by 20.3%, in capsaicin group compared to control group, respectively. (Zhang ZK, Guo X, Lao JQ, Lin YX. Effect of capsaicin-sensitive sensory neurons on bone architecture and mechanical properties in the rat hindlimb suspension model. J Orthop Translat. 2017 Mar 27;10:12-17. doi: 10.1016/j.jot.2017.03.001. eCollection 2017 Jul.)

Other studies also showed the same results. (Offley SC, Guo TZ, Wei T, Clark JD, Vogel H, Lindsey DP, Jacobs CR, Yao W, Lane NE, Kingery WS. Capsaicin-sensitive sensory neurons contribute to the maintenance of trabecular bone integrity. J Bone Miner Res. 2005 Feb;20(2):257-67. Epub 2004 Nov 16)(Ding Y, Arai M, Kondo H, Togari. Effects of capsaicin-induced sensory denervation on bone metabolism in adult rats. Bone. 2010 Jun;46(6):1591-6. doi: 10.1016/j.bone.2010.02.022. Epub 2010 Mar 1)

We think previous studies could prove the question that capsaicin could affect bone metabolism without implant placement. So, in this study, we didn't add this group.

And the same time, we have refined the methods and materials of this manuscript to make this manuscript more reasonable.

3. The microCT scanning parts is not clear. What kind of energy filter was used for scanning this high dense implant inside the bone tissue? And what is the basic imaging processing dealing with metal artefacts and beam hardening?

Answer: The filter we used is CU 0.1mm, the energy/Intensity we used is 90 kVp, 200uA and 18w. The way we used to reduce artifacts is to increase the scan voltage to 90 kVp and increase the integration time to 500ms.

4. The first image in the histological graph was a bit dark compared to other ones.

Answer: Yes, we have changed the first image in the histological graph to make it lighter and similar to others.

5. In the discussion part, " One week later, the corresponding sensory neurons were completely lost", No reference? no info about how to examine.

Answer: After we read other articles, we think that "One week later, the corresponding sensory neurons were completely lost" is not rigorous. So, we think it would be scientific to say " One week later, the capsaicin worked and the selective lesions in sensory fibers happened [7,31]". And the reference is 7 and 31.

6. In the discussion part, "Neurotrophin such as NGF secreted by the local bone could induce the reinnervation of other lower extremity sensory neurons, so the trabecular number and the positive fibers of CGRP-IR of the sensory-denervated group were increased at six weeks compared to three weeks after implantation. Although Neurotrophin increased at six weeks, the trabecular number and the positive fibers of CGRP-IR in the sensory-denervated group still less than that in the sensory-intact group." positive fibers of CGRP-IR was not clear in the previous results part.

Answer: At first, we put the result of CGRP-IR (Abcam, USA) staining in the article. But, after we discussed the manuscript, we deleted the result of CGRP-IR staining, and we didn't find we have discussed the CGRP-IR in the discussion. And now, we think we should put the result of CGRP-IR staining in the supplements. And we have added this in the end of the manuscript. Supplement 1.

Histological images of the femoral cortical bone 3 and 6 weeks after implantation (A: the sensory-intact group of 3 weeks after implantation, B: the sensory-denervated group of 3 weeks after implantation, C: the sensory-intact group of 6 weeks after implantation, D: the sensory-denervated group of 6 weeks after implantation ; CGRP-IR staining, original magnification×100), Black arrows indicates CGRP-IR positive fiber.

7. "Offley et al reported " should be " offley et al. reported".

Answer: Thank you for your kind suggestion. We have changed "Offley et al reported " as " Offley et al. reported" in the third paragraph of discussion.

8. "On the other hand, the Neurotrophin could promote angiogenesis, which would benefit the reconstruction of bone around the implant [38-40]." is this a clear conclusion or just an inference ?

Answer: In Wang's study, she found that a reduced angiogenesis alongside a decline in bone-implant contact percentage and bone mass in a-CGRP -/- mice. Overexpression of aCGRP could partly rescue the impairment. They also showed aCGRP increased vascular volume fraction and mean vessel size, as well as spatially relocated vessels approximate to the region of bone formation. And angiogenic and osteogenic genes were significantly upregulated in the transfection and aCGRP β /p group.

So we infer that "the Neurotrophin could promote angiogenesis, which would benefit the reconstruction of bone around the implant.", and we think it is more reasonable to use "might replace "would".

And this sentence changed as " On the other hand, the Neurotrophin could promote angiogenesis, which might benefit the reconstruction of bone around the implant [38-40]."

Reviewer: 2

Comments to the Author(s)

Please clarify the following questions:

1. There are flaws in language which needs further proofreading and correction.

Answer: We are not native speakers of English, so it is a little hard for us to have a well written in the first time. But we have re-edited the language one sentence by one sentence. After that, we have invited the professionals (Recommended by Associate Editor) to modify the language of the manuscript. Thank you for your suggestions.

2. The authors claim that one week after the injection of capsaicin, the sensory neurons were completely destroyed, for which no tests were carried however. This is theoretically true only when the procedure is properly done and thus a test of sensory function before the placement of implants would be helpful to confirm the sensory denervation.

Answer: Many studies showed that the selective lesions in sensory fibers happened after the injection of capsaicin in one week. And the reference is 7 and 31(Ding Y, Arai M, Kondo H, et al. Effects of capsaicin-induced sensory denervation on bone metabolism in adult rats. *Bone*. 2010; 46: 1591-6. Zhang ZK, Guo X, Lao JQ, Lin YX. Effect of capsaicin-sensitive sensory neurons on bone architecture and mechanical properties in the rat hindlimb suspension model. *J Orthop Translat*. 2017 Mar 27;10:12-17. doi: 10.1016/j.jot.2017.03.001. eCollection 2017 Jul.)

And we have added the CGRP-IR staining in the supplement 1 to state the problem.

3. The grouping and time points of examinations were not clearly described. It was only stated in line 6 of page 3 that there are 40 rats which were divided into two groups. The authors however did several tests with the samples drawn from both femurs of all animals at two time points after the surgery. It was not clarified how many samples from either group underwent each of the examinations, and it is thus not possible to appreciate the correctness of statistics.

Answer: 40 rats were divided in two groups, 20 in each group. 80 implants were used, and one implant was used in left leg and one in right leg for each rat (This information has added in 2.4 Implant surgery). 10 animals were killed at each time point in each group, and 4 parallel tests were carried out for each test. And “N=4” has been added in illustration of each figure.

Appendix B

Dear editor,

We have modified the **Ethics, Data accessibility, Authors' contributions, Competing interests, Acknowledgments and Funding** according to the template requirements. And we have changed it in the manuscript.

Reviewer: 1

Comments to the Author(s)

Thank you for your further work, which makes sense and contributes to the final paper! Please note that it is better to use same term in one paper in stead of using many different ones, eg. micro-CT, μ CT, μ -CT, micro computed tomography.

Answer: I have use the same term in this paper. And I have changed "micro-CT, μ CT, μ -CT, micro computed tomography" to be "micro-CT" all through the paper.